# The Insights into Mitochondrial Genomes of Sunflowers

**DOI:** 10.3390/plants10091774

**Published:** 2021-08-26

**Authors:** Maksim S. Makarenko, Denis O. Omelchenko, Alexander V. Usatov, Vera A. Gavrilova

**Affiliations:** 1The Laboratory of Plant Genomics, The Institute for Information Transmission Problems, 127051 Moscow, Russia; omdeno@gmail.com; 2The Department of Genetics, Southern Federal University, 344006 Rostov-on-Don, Russia; usatova@mail.ru; 3Oil and Fiber Crops Genetic Resources Department, The N.I. Vavilov All-Russian Institute of Plant Genetic Resources, 190031 Saint Petersburg, Russia; v.gavrilova@vir.nw.ru

**Keywords:** perennial sunflowers, mitogenome, *H. occidentalis*, *H. tuberosus*, mitochondrial plasmid

## Abstract

The significant difference in the mtDNA size and structure with simultaneous slow evolving genes makes the mitochondrial genome paradoxical among all three DNA carriers in the plant cell. Such features make mitochondrial genome investigations of particular interest. The genus *Helianthus* is a diverse taxonomic group, including at least two economically valuable species—common sunflower (*H. annuus*) and Jerusalem artichoke (*H. tuberosus*). The successful investigation of the sunflower nuclear genome provided insights into some genomics aspects and significantly intensified sunflower genetic studies. However, the investigations of organelles’ genetic information in *Helianthus*, especially devoted to mitochondrial genomics, are presented by limited studies. Using NGS sequencing, we assembled the complete mitochondrial genomes for *H. occidentalis* (281,175 bp) and *H. tuberosus* (281,287 bp) in the current investigation. Besides the master circle chromosome, in the case of *H. tuberosus*, the 1361 bp circular plasmid was identified. The mitochondrial gene content was found to be identical for both sunflower species, counting 32 protein-coding genes, 3 rRNA, 23 tRNA genes, and 18 ORFs. The comparative analysis between perennial sunflowers revealed common and polymorphic SSR and SNPs. Comparison of perennial sunflowers with *H. annuus* allowed us to establish similar rearrangements in mitogenomes, which have possibly been inherited from a common ancestor after the divergence of annual and perennial sunflower species. It is notable that *H. occidentalis* and *H. tuberosus* mitogenomes are much more similar to *H. strumosus* than *H. grosseserratus*.

## 1. Introduction

In contrast to animals, in which mitochondrial genomes are usually conserved in size and gene content across large taxonomic groups, plant’s mitogenomes display high variability in size and structure, as well as having distinct features even in closely related species [1,2]. The angiosperm mitogenome size ranges from 66 kbp in *Viscum scurruloideum* [3] to 11.3 Mbp in *Silene conica* [4]. Such a difference in higher plant mitochondrial DNA (mtDNA) may be associated with frequent insertions, rapid rearrangements, and complex multipartite structures, often involved in recombination [5]. Despite considerable variations in mitogenome size, a large fraction of mitochondrial genes are stable in content and show very low sequence divergence [6,7]. The significant difference in the mtDNA size and structure with simultaneous slow evolving genes makes the mitochondrial genome paradoxical among all three DNA carriers in the plant cell. Such features make the mitochondrial genome interesting to study [8].

Sunflowers are members of Asteraceae, the largest family of flowering plants. The *Heliantheae* is one of the largest of its tribes, and it has well over 2000 species [9]. Sunflowers occupy diverse habitats, including various disturbed or extreme habitats [10]; have high evolution rates [11]; and have a significant variation in ploidy level and genome length [12]. All these features make the *Helianthus* genus members exciting for fundamental research. On the other hand, there are at least two economically valuable species in the *Helianthus* genus: common sunflower (*H. annuus* L.) and Jerusalem artichoke (*H. tuberosus*). Common sunflower (*H. annuus* L.) is a globally important oilseed, food, and ornamental crop, the second-largest hybrid crop, and the fourth largest oilseed crop worldwide (FAO, 2019). At the same time, the Jerusalem artichoke is a crop of great potential for food, production of biofuels, and industrial products [13]. Thus, in turn, it elevates the practical importance of *Helianthus* genus investigations.

The successful investigation of the sunflower nuclear genome [14] provided insights into some genomics aspects and significantly intensified sunflower genetic studies [10,15,16,17]. However, the investigations of organelles’ genetic information in *Helianthus*, especially devoted to mitochondrial genomics, are presented by limited studies [18,19]. To date, only three complete mitogenomes of *Helianthus* species, including single annual species, are available—*H. annuus* [20] and two perennial species: *H. grosseserratus* and *H. strumosus* [19]. Such limited data are not sufficient for estimating patterns of mitochondrial genome features in the *Helianthus* genus.

Mitochondrial genome investigations in sunflowers also have an important practical issue since mtDNA is associated with the cytoplasmic male sterility (CMS) phenotype [21]. CMS is one of the most beneficial mutations that allows for heterotic hybrid production [22]. CMS sources are often obtained through inter- or intraspecific hybridization [23]. Many sunflower species, mostly diploid ones, have been used to develop new CMS sources for hybrid breeding in sunflower [23]. The field trials with the perennial sunflower species mentioned high levels of interspecies fertility between *H. annuus* and *H. tuberosus* [22,24,25].

In this work, we assembled the complete mitogenomes of two perennial sunflower species: *H. occidentalis* and *H. tuberosus*. We studied their gene profile and analyzed sequence and structure in comparison to currently available complete sunflower mitogenomes.

## 2. Results

After reads trimming, 1,891,747 (*H. occidentalis*) and 2,283,878 (*H. tuberosus*) paired reads were gained. Since the content of mitochondrial DNA (more than 30% of total reads) was high, only a few high-coverage (>500 depth) contigs were generated, which made genome assembly easier. As a result, master cycle mitochondrial chromosomes were obtained for both studied sunflower species. The complete mitogenome of *H. occidentalis* (Figure 1A) counts 281,175 bp, and *H. tuberosus* (Figure 1B)—281,287 bp, which makes them more than 19.5 kbp smaller than the mitogenome of *H. annuus.* However, their mitogenomes have a length similar to other perennial (*H. grosesserratus*, *H. strumosus*) sunflower species. The GC content in both studied species is almost the same: 45.23% (*H. occidentalis)* and 45.22% (*H. tuberosus*). Besides the master cycle form, the mitogenome of each species can be presented by at least three subgenome circles (Figure 1) formed by the 150–250 bp repeats.

The comparative analysis of *H. tuberosus* and *H. occidentalis* mitogenomes has pointed out the great similarity in their sequences. No significant (>100 bp) insertions or deletions were found. According to full-length mtDNA alignment, the nine syntenic blocks were defined (Figure 2). Many more rearrangements can be noticed while comparing mitogenomes of *H. occidentalis* and *H. tuberosus* with other annual (*H. annuus*) and perennial (*H. grosesserratus*, *H. strumosus*) sunflower species (Appendix A).

We found five long deletions (>200 bp) in *H. occidentalis* and *H. tuberosus* mtDNA compared to *H. annuus*: 0.44 kbp, 3.18 kbp, 4.1 kbp, 6.26 kbp, and 14.44 kbp. Regardless of significant deletion sizes, only copies of two tRNA genes (*trnI*, *trnK*) are missing in the mitogenomes due to 14.44 kbp deletion, while the other deletions are located in intergenic regions (IGR) and thus do not affect mtDNA coding sequences at all. The most significant deletion (14.44 kbp) predominantly affects the large repeat region (12.93 kbp) of the annual sunflower mtDNA *cox2–nad2* (286,492–300,787 positions in MN171345.1). The identical deletion is observed in *H. strumosus* mitogenome, while in the case of *H. grosseserratus*, there is only a partial deletion of a large repeat region (7.98 of 12.93 kbp). The smallest deletion (0.44 kbp) was also found in *H. strumosus*, but not in *H. grosseserratus*. Putting the 3.18 kbp, 4.1 kbp, and 6.26 kbp deletions on a genetic map of *H. annuus*, we found that they have the following IGR localization: *atp6-cox2*, *cob-ccmFC*, and *nad4L-orf259*, respectively. Deletions of the same or similar size are also present in these IGRs in other perennial species (*H. strumosus* and *H. grosseserratus*).

Two insertions (>200 bp) were detected in *H. occidentalis* and *H. tuberosus* mtDNA while comparing with *H. annuus*. The first one is a 4.7 kbp insertion. *H. strumosus* has a similar insertion, but it counts 3.2 instead of 4.7 kbp. The rest of the insertion (1.5 kbp) includes *orf300*, specific for H. occidentalis and *H. tuberosus* (Figure 3). The *orf300* has 99% similarity to hypothetical protein (NCBI ID KAF5771886.1), which was annotated in the 13 chromosome of *H. annuus*. This insertion seems to originate from nuclear DNA due to the transfer of genetic material between the nucleus and mitochondrion. Another insertion (1.99 kbp) is also a result of genetic exchange, but between plastid and mitochondrion. It has 100% similarity to the part of the inverted repeat sequence of many annual and perennial sunflower chloroplast DNA. Exactly the same insertion was discovered in HA89 (MAX1) sunflower line with cytoplasm initially obtained from *H. maximiliani* [26]. Such insertion has also been detected in *H. strumosus*, but not in *H. grosseserratus*.

One of the most notable features we have discovered is the mitochondrial circular plasmid in the case of *H. tuberosus.* The 1488 bp contig, including third and fifth self-complement regions equal to 127 bp (k-mer length), with more than fivefold higher coverage than other mitochondrial contigs, allowed us to annotate the 1361 bp sequence as a mitochondrial circular plasmid. The remapping of mitochondrial reads we obtained in previous studies for such species as *H. grosesserratus* and *H. strumosus* gained no result; as for *H. annuus* reads, they mapped to 950–1250 region of the plasmid. Exactly this 300 bp region has similarity with pIT mitochondrial plasmid (NCBI accession M36422.1) and plasmid-like DNA (P1) repeat sequences (NCBI accession X15844.1), which have been identified in *H. annuus* [27,28]. It is of note that the most part (about 800 bp) of the discovered mitochondrial plasmid has an identity to predicted ncRNA (NCBI accession XR_004872305.1).

The mitochondrial gene content is identical for both sunflower species, counting 32 protein-coding genes, 3 rRNA, and 23 tRNA genes. In the studied mitogenomes, we annotated protein-coding genes, including NADH:ubiquinone oxidoreductase genes (*nad1*,−*2*,−*3*,−*4*,−*4L*,−*5*,−*6*,−*7*,−*9*), succinate dehydrogenase gene *sdh4*, cytochrome c oxidase genes (*cox1*,−*2*,−*3*), ATP synthase genes (*atp1*,−*4*,−*6*,−*8*,−*9*), cytochrome c biogenesis genes (*ccmB*,−*C*,−*FC*,−*FN*), ribosomal protein genes (*rps3*,−*4*,−*12*,−*13*, *rpl−5*,−*10*,−*16*), *cob*, *matR*, and *mttB*. The annotated tRNA genes can be divided into two groups, typical mitochondrial and chloroplast-like genes. There are 7 chloroplast-like genes and 16 (12 unique) native mitochondrial tRNA genes. Some genes (*trnK*, *trnM*, *trnN*, *trnQ*) are presented in several copies, and thus, in summary, all identified tRNA genes are encoding transfer molecules for 17 amino acids. We also annotated open reading frames by several criteria: (1) similarity to other mitochondrial proteins or presence of some defined protein domains (2) location close to other mitochondrial genes and thus a possibility to be co-transcripted in polycistronic mRNA. Thus the 18 ORFs have been annotated in both studied mitogenomes: *orf117*, *orf126*, *orf139*, *orf148*, *orf148B*, *orf156*, *orf161*, *orf163*, *orf184*, *orf207*, *orf222*, *orf270*, *orf291*, *orf295*, *orf298*, *orf300*, *orf316*, *orf365*. The mitochondrial gene and ORF content for annual (*H. annuus*) and four perennial species are summarized in Figure 3.

Comparative analysis of mtDNA sequences between four perennial species revealed 6 SNPs that are unique for *H. tuberosus* and 12 SNPs for *H. occidentalis.* One polymorphic site was found to be common for *H. occidentalis* and *H. tuberosus* while comparing them with *H. grosesserratus* and *H. strumosus* (Appendix A). Noticeably, all discovered SNPs are located in the intergenic regions.

A total of 18, 19, 20, and 19 SSRs were found in mitochondrial genomes of *H. occidentalis*, *H. tuberosus*, *H. strumosus*, and *H. grosseserratus*, respectively. The most abundant SSRs in all analyzed species are mono- and di-nucleotide motifs (33–47% each of all SSRs); the next most frequent motif is tetra-nucleotide (15–20%); while tri-, penta-, and hexanucleotide motifs have been found in significantly lower numbers, or even absent in some species (Table 1). Among the most abundant SSRs, A/T-containing motifs are most frequent (Appendix A). This distribution is consistent with the known domination of A/T mono- and di-nucleotide SSRs in plant mitochondrial genomes [29]. Comparative analysis of SSRs and their flanking sequences between species has shown that *H. occidentalis*, *H. tuberosus*, and *H. strumosus* share seven SSRs (almost half of all found SSR for each species). At the same time, almost all SSRs of the *H. grosseserratus* mitogenome (15 out of 19) are specific to this species (Appendix A), demonstrating its phylogenetic distance from other investigated species. Among shared SSRs, five are polymorphic mono-nucleotide SSRs: two between *H. tuberosus* and *H. grosseserratus*, two between *H. occidentalis* and *H. strumosus*, and one between *H. occidentalis* and *H. grosseserratus*. Only one of them is a length variation of polyG SSR (*H. occidentalis* and *H. strumosus*), while the others are polyT. All the SSRs found are in intergenic regions, except for one SSR inside *nad7* intron of both *H. occidentalis* and *H. tuberosus*.

## 3. Discussion

The comparison of complete mitochondrial genomes revealed that *H. occidentalis* and *H. tuberosus* have almost identical DNA sequences without significant deletions and insertions (deletions/insertions < 100 bp, few SNP); both species have common genes and ORFs content. However, there are some rearrangements in syntenic blocks’ order. Master circle is often not the predominant form of plant mtDNA [30], and dynamic homologous recombination between mitochondrial subgenomes may result in different master circles [8,31,32], which could be the cause of observed rearrangements. While comparing mitogenomes of annual and perennial sunflowers (Appendix A), we identified several rearrangements. Most of the detected deletions and insertions have similar sequences and are localized in the same mtDNA regions of the studied perennial species. It is possible that such changes were inherited from a common ancestor after the divergence of annual and perennial sunflower species. However, in perennial sunflower comparison, *H. occidentalis* and *H. tuberosus* mitogenomes are much more similar to *H. strumosus* than *H. grosseserratus*.

The studied mitochondrial genomes (*H. occidentalis*, *H. tuberosus*) have typical gene content for mitochondria of a flowering plant [33,34], with 32 protein-coding genes. The same gene content was discovered in other perennial species (H. strumosus), while *H. annuus* and *H. grosseseratus* have 31 protein-coding genes since sdh4 is presented as a pseudogene (Figure 3) in their mitogenomes. It is notable that all investigated sunflower mitogenomes lack the *sdh3* gene. The succinate dehydrogenase genes (*sdh3*, *sdh4*) are among the “unstable” genes in mitogenome, and their presence is highly variable, even among evolutionary close angiosperm species [33,35]. Other genes that can be often excluded from mitogenome, for instance, by transfer to the nucleus, are ribosomal (*rpl*, *rps*) genes [36]. Basal angiosperm species have 14–15 ribosomal genes in mitochondrial genomes [33,37], and in the studied sunflowers, only seven ribosomal genes have been annotated. Such content of ribosomal genes is common for Asteraceae mitogenomes [38]. Nevertheless, sunflowers lack *rps1*, *-14*, and *-19* genes, presented in some Asteraceae [38]. Among other ribosomal genes, *rpl2* and *rps10* are absent in mitochondria of many plant lineages [33,34,35], being found transferred to the nucleus [36]. At the same time, lack of *rps2*, *rps7*, and *rps11* may be a typical feature of Asteraceae mitogenomes, or at least a genus close to *Helianthus*.

The existence of species-specific mitochondrial plasmids in plants has been established in different studies [39,40]. However, there is still little information about the biological roles of mitochondrial plasmids and their appearance in the mitochondrion [41,42]. Though the number of mitochondrial genome investigations using NGS techniques has significantly increased [43,44,45], we could not find reports about mitochondrial plasmids in such studies to the best of our knowledge. In case of similarity between mitochondrial plasmids and master chromosomes and short-read sequencing (<300 bp), the plasmid sequences could be integrated into the mtDNA during assembly. While using long reads (ONP, PacBio), such plasmids (less than 3–4 kbp) may also be missed, for instance, in the process of contigs filtration. However, several features allowed us to identify the plasmid in the current study: preliminary mitochondrial fraction isolation, no similarity between the plasmid and the main mtDNA sequence, and high stoichiometry relative to the mitogenome. Interestingly, the discovered circular plasmid has great similarity to long non-coding RNA (lncRNA) from the nuclear genome of *H. annuus*. Accumulating evidence indicates that a large fraction of lncRNAs are located in the cytosol, and involved in different processes and signaling pathways in organelles, including those regulating the metabolism in mitochondria [46,47], and may influence mitochondrial–nuclear interactions [48]. The significance of the similarity between sequences of *H. annuus* lncRNA and discovered in *H. tuberosus* mitochondrial plasmid is unclear but seems to be not accidental.

Distant hybridization leads to an imbalance in the regulatory processes of the cell [49], including those participating in nuclear–mitochondrial relationships. Thus, significant changes in mitogenomes of hybrids may be observed: high rates of rearrangements or recombination [50] and even shift of the inheritance pattern [51]. Therefore, the investigations of mtDNA of interspecific hybrids and comparison with that of their parents are of particular relevance. The interspecific hybrids between *H. annuus* and perenial species (including *H. occidentalis*, *H. tuberosus*, and *H. strumosus*) were obtained by employees of the N.I. Vavilov All Russian Institute of Plant Genetic Resources. In the current study, we analyzed the mitogenomes of parent forms, and future research will be devoted to the mtDNA of the hybrids. Such investigations can be beneficial for understanding the mechanisms of mtDNA reorganizations, which are paradoxically common in plant mitogenomes [52]. Moreover, interspecific hybridization often results in obtaining new CMS sources. Finally, mitogenomic research can help to explore the molecular processes leading to CMS phenotype formation. It is important to note that besides CMS phenotype, some other agronomic traits can be associated with the mitochondrial genome. For instance, in rapeseed, elevated seed oil content is determined by mitochondrial *orf188* [53]. In the case of investigated sunflower species, mtDNA sequences similar to rapeseed *orf188* were not detected. However, the role of discovered ORFs in *Helianthus* species mitogenomes is underexplored. Since more mitogenomic data become available each year, future studies will provide more insights into these unresolved questions.

## 4. Materials and Methods

### 4.1. Plant Material and Mitochondrial DNA Extraction

The *H. occidentalis* and *H. tuberosus* sunflower species were obtained from the genetic collection of the N. I. Vavilov All-Russian Institute of Plant Genetic Resources, Saint Petersburg, Russia. Plant leaves were used for mitochondrial DNA isolation. For this purpose, 5 g of leaves (without petiole and midrib) were homogenized by mortar and pestle in 20 mL of STE buffer (0.4 M sucrose, 50 mM Tris (pH 7.8), 4 mM EDTA-Na2, 0.2% bovine serum albumin, 0.2% 2-mercaptoethanol), filtered twice with 100-micron mesh and then centrifuged using several steps: (1) 500× *g* for 5 min, picking the supernatant; (2) 3000× *g* for 5 min, picking the supernatant; (3) 12,000× *g* for 15 min, discarding the supernatant. The pellet was treated using 10 units of DNAse (Syntol, Moscow, Russia) for 7 min and then used for DNA isolation. For both samples, the DNA extraction was performed with the PhytoSorb kit (Syntol, Moscow, Russia), according to the manufacturer’s protocol. The DNA concentration was measured with a Qubit 4 fluorometer (Thermo Fisher Scientific, Waltham, MA, USA).

### 4.2. Next-Generation Sequencing

A total of 10–20 ng of extracted DNA was fragmented using a Covaris S220 sonicator (Covaris, Woburn, MA, USA). Then, NGS libraries were prepared with NEBNext Ultra II DNA Library Prep Kit for Illumina (New England Biolabs, Ipswich, MA, USA), following the manufacturer’s guidelines and using 14 PCR cycles. The fragment length distribution of the prepared libraries was determined with Bioanalyzer 2100 (Agilent, Santa Clara, CA, USA), and the concentrations were evaluated with Qubit 4 fluorometer (Thermo Fisher Scientific, Waltham, MA, USA) and qPCR. The NGS libraries were diluted to 10 pM and then sequenced on MiSeq (Illumina, San Diego, CA, USA) with MiSeq Reagent Kit v2 (500 cycles). We generated 2.3 and 2.9 million 250 bp paired reads for *H. occidentalis* and *H. tuberosus*, respectively (deposited to SRA under BioProject ID PRJNA731105).

### 4.3. Mitochondrial Genome Assembly and Annotation

For quality control of reads, we used FastQC v0.11.9 (https://www.bioinformatics.babraham.ac.uk/projects/fastqc/, accessed on 12 April 2021). Trimmomatic v0.39 software [54] with several options (ILLUMINACLIP, SLIDINGWINDOW:5:15, MINLEN:65) was used to trim adapters and discard short or low-quality reads. Contigs were generated with SPAdes Genome Assembler v3.13.1 [55] using 127 k-mer and cut-off threshold equal 70. The whole mitochondrial genome assemblies were based on high coverage (>500 depth) contigs, selected using Bandage v0.8.1 [56] program for visualizing de novo assembly graphs. The genome assemblies were validated by remapping reads with Bowtie 2 v2.3.5.1 [57] and visual revision of coverage uniformity (especially in the junctions of contigs) using Tablet v1.19.09.03 [58]. The mitochondrial genomes were annotated with GeSeq [59] and previously gained mitochondrial genome of *H. strumosus* (MT588181.1 NCBI accession). The ORFfinder (https://www.ncbi.nlm.nih.gov/orffinder, accessed on 1 May 2021) was used to discover new open reading frames (ORFs). The potential ORFs were analyzed using BLAST tool [60] and InterProScan tool (https://www.ebi.ac.uk/interpro/search/sequence/ accessed on 20 May 2021). Graphical genome maps were generated using the Circos tool v0.69-9 [61]. The whole-genome alignments were performed with Mauve tool v2.4.0 [62].

### 4.4. Detection and Analysis of SSR

SSR loci were identified by SSRMMD tool [63] using simple regular expression mode and 10, 6, 5, 4, 4, and 4 repeat number thresholds for mono, di-, tri-, tetra-, penta-, and hexanucleotide motifs, respectively. Common and polymorphic SSR loci were identified pairwise between mitogenomes by conservativeness of 100 bp SSR flanking sequences scored with Needleman–Wunsch alignment. The SSR detection results were collected, combined, and drawn using custom python scripts and pyvenn library. The search for the intersection of SSR with annotated features of mitogenomes was performed using BEDTools v2.30.0 [64].

## 5. Conclusions

The complete master cycle mitochondrial chromosomes were obtained for *H. occidentalis* (281,175 bp) and *H. tuberosus* (281,287 bp). The mitochondrial gene content is identical for both sunflower species, counting 32 protein-coding genes, 3 rRNA, 23 tRNA genes, and 18 ORFs. The 1361 bp circular plasmid was revealed in *H. tuberosus* mitochondrial genome. The specific SSR and SNPs were identified. Comparison of perennial sunflowers with *H. annuus* allowed us to establish common rearrangements in mitogenomes, which have possibly been inherited from a common ancestor after the divergence of annual and perennial sunflower species. However, in the comparison of perennial sunflowers, we found that *H. occidentalis* and *H. tuberosus* mitogenomes are much more similar to *H. strumosus* than *H. grosseserratus*.

## Figures and Tables

**Figure 1 plants-10-01774-f001:**
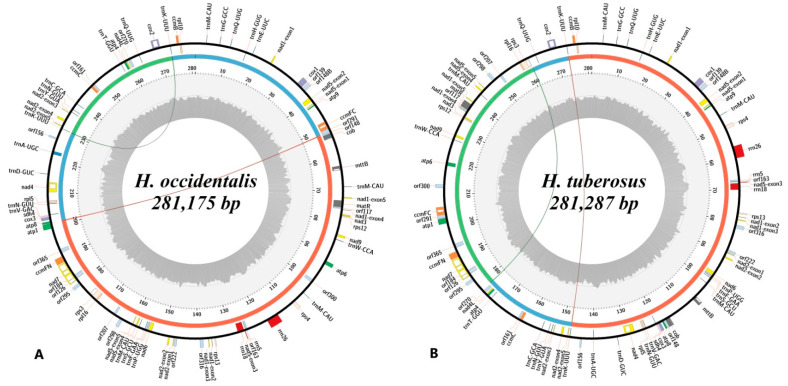
Genetic maps of the *H. occidentalis* (**A**) and *H. tuberosus* (**B**) mitogenomes. Colored blocks on circular axis denote genes, and white blocks denote introns. The gray histogram on the inner ring shows the guanine-cytosine (GC) content as percent of GC nucleotides in 100 bp sliding window, the dark gray line denotes 50% GC content threshold. Arcs indicate repeats associated with structural variants that could form subrings indicated by arcs and red, blue, and green regions on the middle ring of the figure.

**Figure 2 plants-10-01774-f002:**
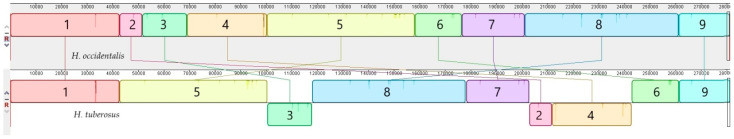
The progressive MAUVE alignment of *H. occidentalis* and *H. tuberosus* complete mitogenomes. According to sequence similarity, 9 syntenic blocks with a different order were identified.

**Figure 3 plants-10-01774-f003:**
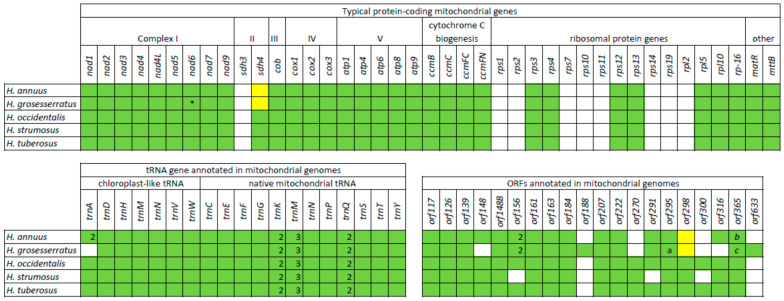
The distribution of protein-coding, tRNA genes, and ORFs among *Helianthus* species with complete mitogenomes. White square—the absence of the gene, green square—the presence of the gene, yellow square—the presence of pseudogene (premature stop codon). *, a, b, c = C–terminus shortening of the encoded proteins: a = *orf216*, b = *orf334*, c = *orf284*.

**Table 1 plants-10-01774-t001:** SSR motif count.

Motif Length (bp)	*H. occidentalis*		*H. tuberosus*		*H. strumosus*		*H. grosseserratus*	
	SSR Number	%	SSR Number	%	SSR Number	%	SSR Number	%
Mono-	6	33.3	7	36.8	8	40	9	47.4
Di-	7	38.9	7	36.8	7	35	7	36.8
Tri-	1	5.6	1	5.3	0	0	0	0.0
Tetra-	3	16.7	3	15.8	4	20	3	15.8
Penta-	1	5.6	1	5.3	1	5	0	0.0
Hexa-	0	0.0	0	0.0	0	0	0	0.0
Total	18	100.0	19	100.0	20	100	19	100.0

## Data Availability

The raw NGS reads were deposited to NCBI SRA under BioProject ID PRJNA731105. The complete mitochondrial genomes were deposited to NCBI with following GenBank accession numbers MZ147621 (*H. occidentalis*) and MZ147622 (*H. tuberosus*).

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
