# Peer review of "The Insights into Mitochondrial Genomes of Sunflowers"

_plants, 2021, doi:10.3390/plants10091774_

Round 1

Reviewer 1 Report

In this submission, Makarenko and colleagues describe sequencing, assembly, and annotation of mitochondrial genomes from two perennial sunflower species, Helianthus occidentalis and H. tuberosus. The authors present a comparative analysis of the mitochondrial genomes for these two species along with previously sequenced mitochondrial genomes for the common annual sunflower H. annuus and two other perennial species, H. grosseserratus and H. strumosus. The experimental work is appropriate and sufficiently well-described. I do think that the authors could provide some additional discussion of the novelty and/or importance of the work. 

Specific comments:

  1. Why have the authors selected H. tuberosus and H. occidentalis for this study? Was there any reason to choose these two species? The manuscript would probably benefit from a supplemental figure showing the  phylogenetic relationships among the species compared in this study.
  2. The authors describe differences among the mitochondrial genomes examined, but provide relatively little discussion about why it is important to study this topic. The authors briefly mention that polymorphisms identified could be used for "molecular barcoding." In what context? What would be the value of this? In general, the authors should provide additional discussion of the value or utility of the information gained in the study.
  3. Related to the above comment, what do the authors see as future work needed in this area? Would it be useful to sequence and compare additional mitochondrial genomes from the Heliantheae or Asteraceae, or does the current set provide a sufficient representation? 

Author Response

We are grateful for suggestions and comments on the paper. It allowed us to improve the manuscript.

The discussion section was revised. We added the information highlighting the importance of the work. 

  1. Several obstacles influenced our choice of these species. H. tuberosus is an agronomically valuable (crop) species of Helianthus genus. In contrast, H. occidentalis is one of the most unstudied Helianthus species. We also selected these species since we plan to investigate mitogenomes of interspecific hybrids H. annuus X H. tuberosus and H. annuus X H. occidentalis (obtained in N. I. Vavilov All-Russian Institute of Plant Genetic Resources). The phylogenetic tree, according to today's mitogenomic data, is uncertain. There are too low rates of substitutions in mitochondrial genes of the studied species, thus leading to low bootstrap values. So we did not include the phylogenetic tree at all. We plan to show phylogenetic relationships among the Helianthus species in future studies with more data. However, if you insist, we can add the cladogram based on the mitochondrial genes alignment in the Supplemental data. 
  2. The discussion was edited. In general, all genomic polymorphisms which can be unique for some specific species can be used for molecular barcoding, but it makes no great sense. While we have information about the mtDNA sequence of only 5 of 50 species in the genus, it is unreliable to make statements about "barcoding". Therefore the phrases which mentioned "molecular barcoding" were excluded. 
  3. The current study provides information about some Helianthus species, which expands knowledge in this field of research, but is not sufficient for making fundamental statements e.g. the estimation of microevolution processes in mitogenomes of sunflowers. Future studies in this area may be devoted to broadening the spectrum of investigated species and evaluating the polymorphisms of mtDNA in populations of sunflowers. For instance, there is quite a bit of NGS data (SRA) for some Helianthus species. Using both current and SRA data, some new studies may be provided. We are planning to continue our research in this field.

Reviewer 2 Report

The manuscript by Makarenko et al.; reports mitochondrial sequencing of the genome of two new members of the Helianthus family. Previously, the authors of the current manuscript analyze in details another member of the Helianthus family.  The importance of the present work is the mitochondrial genome of Jerusalem artichoke, another member of Helianthus. Jerusalem artichoke does not belong to the artichoke family. Importantly to my knowledge, it is only Helianthus species where tubers are consumed as food worldwide. 
Generally, I find the manuscript interesting, but it should include some additional analysis in light of old publications.

-My suggestion is to include phylogenetic analysis of the helianthus family. 

-The absence of Orf188 in some Helianthus species is interesting. Recent work in rapeseeds suggest that in involve in seed oil contents? (https://doi.org/10.1016/j.molp.2019.01.012)

Author Response

We are grateful for your assessment of the article. The comments are valuable. 

1) The phylogenetic tree, according to today's mitogenomic data, is uncertain. There are too low rates of substitutions in mitochondrial genes of the studied species, thus leading to low bootstrap values. So we did not include the phylogenetic tree at all. We plan to show phylogenetic relationships among the Helianthus species in future studies with more data. However, if you insist, we can add the cladogram based on the mitochondrial genes alignment to the Supplemental data of the manuscript. 

2) In the case of investigated sunflower species, no mtDNA sequences similar to rapeseed orf188 were detected. We mentioned it in the new variant of the manuscript. We are grateful to you for providing the reference on such an exciting study, broadening our knowledge.

Round 2

Reviewer 2 Report

The authors addressed points previously raised by the reviewer and improved the manuscript. As a result, the manuscript is more apparent in the present form and encourages future research. 

This manuscript is a resubmission of an earlier submission. The following is a list of the peer review reports and author responses from that submission.